# Do Bryophyte Elemental Concentrations Explain Their Morphological Traits?

**DOI:** 10.3390/plants10081581

**Published:** 2021-07-31

**Authors:** Marcos Fernández-Martínez, Jordi Corbera, Oriol Cano-Rocabayera, Francesc Sabater, Catherine Preece

**Affiliations:** 1Research Group PLECO (Plants and Ecosystems), Department of Biology, University of Antwerp, 2610 Wilrijk, Belgium; catherine.preece09@gmail.com; 2Delegació de la Serralada Litoral Central, ICHN, 08302 Mataró, Catalonia, Spain; corberajordi@gmail.com (J.C.); canorocabayera@gmail.com (O.C.-R.); fsabater@ub.edu (F.S.); 3Department of Ecology, University of Barcelona, 08028 Barcelona, Catalonia, Spain

**Keywords:** stoichiometry, morphology, traits, elemental composition, moss, cell

## Abstract

Differences in the elemental composition of plants, mainly C, N, and P, have been shown to be related to differences in their nutritional status, and their morphological and functional traits. The relationship between morphological traits and micronutrients and trace elements, however, has been much less studied. Additionally, in bryophytes, research devoted to investigating these relationships is still very scarce. Here, we analysed 80 samples from 29 aquatic and semi-aquatic (hygrophytic) moss species living in Mediterranean springs to investigate the relationship between moss nutrient concentrations and their micro- and macroscopic morphological traits and growth forms. We found that, across species, the elemental concentration of mosses was more tightly linked to macroscopic traits than to microscopic traits. Growth forms could also be successfully explained by the concentration of elements in mosses. Apart from macronutrients and their stoichiometric ratios (C:N, C:P, and N:P), micronutrients and trace elements were also important variables predicting moss morphological traits and growth forms. Additionally, our results showed that microscopic traits were well related to macroscopic traits. Overall, our results clearly indicate that the elemental composition of mosses can be used to infer their morphological traits, and that elements other than macronutrients should be taken into account to achieve a good representation of their morphological and, potentially, functional traits when comparing the elemental composition across species.

## 1. Introduction

Bryophytes are amongst the most fascinating organisms of the plant kingdom, given their ecology, physiology, and morphology. Their lack of true cuticles and roots make them very sensitive to environmental conditions [1], and those features make them well suited to monitoring changes in environmental conditions. Particularly under environmental pollution, bryophytes have been shown to accumulate heavy metals and change their elemental composition or, in the event that they cannot tolerate the new conditions, they disappear. Accordingly, bryophytes have been used to assess the ecological consequences of air and water pollution (e.g., nitrate or heavy metals) [2,3,4,5,6,7]. However, their elemental composition under normal conditions, and how it is related to their functioning, have been seldom explored, despite a few studies focused on bryophyte C, N, and P or macronutrient concentrations [8,9,10,11,12]. Nonetheless, information on the elemental composition of bryophytes is particularly lacking for a large array of micronutrients and trace elements, particularly from non-polluted areas.

The elemental composition of organisms has been repeatedly shown to be a very good indicator of their morphological and functional traits (e.g., photosynthesis, respiration), the ecological strategies they follow, and their relationship with their environment [13,14,15,16,17]. Accordingly, the elemental composition of organisms has been repeatedly shown to present an important adaptive value, both in plants and animals [18,19]. Generally, plants presenting higher concentrations of N and P, and lower C:N and N:P ratios, tend to be fast growing and more productive; reproduce more, and more frequently; and require fewer defence mechanisms; while a more conservative lifestyle is shown by nutrient-limited plants [15,20,21,22]. Similar patterns have also been shown in bryophytes [23,24] suggesting that a link between the elemental composition of plants and their morphological and functional traits should occur throughout the plant kingdom.

Recent publications focused on bryophyte functional traits have considerably increased our knowledge regarding the relationship between the environment and bryophyte functional traits [25,26,27,28,29]. In particular, some of these studies identified water chemistry (e.g., pH, dissolved nutrients, and heavy metals) as an important determinant of moss traits and growth forms [2,25,28]. Other studies, focused on how several traits are related to each other, made very relevant contributions in order to emulate the leaf economic spectrum described for vascular plants [23,24]. These studies provided very relevant information on how bryophyte morphological traits, and N and P concentrations are related to their photosynthesis and dark respiration rates, indicating similar patterns to those observed in vascular plants [22].

However, to the best of our knowledge, no efforts have been devoted to investigating whether the elemental composition of bryophytes is related to their micro- and macroscopic morphological traits and growth forms across a large number of species. To fill in this gap, we here analysed 80 samples from 29 aquatic and semi-aquatic (hygrophytic) moss species to investigate the relationship between gametophyte moss nutrient concentrations and their morphological traits. We hypothesised that the elemental composition of moss species will be a very good indicator of their morphological traits, similar to that which has been reported in vascular plants [15,17,22]. Additionally, we hypothesise that moss species presenting higher concentrations of N and P, and low C:N, C:P, and N:P stoichiometric ratios, related to more productive organisms, will present larger and wider leaves and will be lighter (i.e., less dense and less mass per area). Our results will help us to further understand how the elemental composition of organisms determine their functional traits and ecological strategies in bryophytes.

## 2. Methods

### 2.1. Data Collection and Analyses of Elemental Composition and Morphological Traits

We used previously published data in which we analysed the elemental composition of moss samples [30,31] from a large collection used in a previous study [25] for which their morphological traits were already estimated, including 29 different hygrophytic moss species. From the total collection of 100 samples with morphological traits, 80 samples had enough mass to analyse their elemental composition. Those 80 samples were used to perform this study. Moss samples were collected from springs distributed across Catalonia (north-eastern Iberian Peninsula, Appendix A) following a large climate and hydrochemical gradient [32]. In the subset of springs used here, water pH ranged from 5.17 to 8.34 (median: 7.20) and water conductivity ranged from 24 to 2094 (median: 498) µS cm^−1^. A detailed analysis of the water chemical composition can be found in the online material from references [25,30,31]. All mosses collected were in direct contact with the water of the springs throughout the year, with only occasional interruptions in water flow during winter and summer (when water freezes or during intense drought events, respectively). Given that these springs were draining water from aquifers, their water chemical composition was very stable in time. The climate of the area of study is Mediterranean, ranging from humid to sub-humid, and with large differences in mean annual temperature and precipitation (from 4.2 to 15.7 °C, and from 567.4 to 1202.4 mm y^−1^, respectively) [25,33].

Before the moss elemental analyses, we submerged all samples in a solution of acetic acid at pH 2.7 for 15 min in order to remove incrustations of CaCO_3_. A few samples required up to an hour to remove all CaCO_3_ incrustations. We then rinsed the samples with distilled water and dried them at 60 °C for 48 h. We ground the samples to a powder using liquid nitrogen and a mortar. Moss C and N concentration and δ^13^C and δ^15^N were determined by using a Flash EA1112 and TC/EA coupled to a stable isotope mass spectrometer Delta C through a Conflo III interface (ThermoFinnigan). We determined the concentration of 19 elements, including macro-, micronutrients and trace elements (P, K, S, Ca, Mg, Na, B, Fe, As, Al, Cd, Co, Cu, Cr, Hg, Mn, Ni, Pb, and Zn) by inductively coupled plasma mass/optical emission spectrophotometry (ELAN 600 and Optima 8300, respectively, Perkin Elmer) after the samples were digested overnight at 90 °C with nitric acid and hydrogen peroxide in a 2:1 ratio. All analyses were carried out by the technical staff at the Scientific and Technical Centres of the University of Barcelona. We additionally calculated the C:N, C:P, and N:P stoichiometric ratios on a mass basis for further analyses.

We used seven macroscopic morphological moss traits estimated for [25], including leaf length, width, form and area, water absorption capacity (WAC), moss density (moss dry mass per volume), and moss mass per area (MMA). Measurements of leaf length, width, and area were then taken by measuring three leaves per sample placed in a coverslip and using *ImageJ* software over the pictures taken with a microscope. Leaf form was then calculated as the leaf length-to-width ratio. Moss mass per area was calculated as the ratio of moss dry mass (determined with a precision scale) to projected area (through photographs of samples on top of graph paper and using *ImageJ* to calculate their area). Water absorption capacity was determined by calculating the fresh-to-dry weight ratio of the same moss sample used to calculate MMA. We rehydrated the samples by submerging them into distilled water for three minutes in a graduated cylinder of 10 mL. We then recorded the volume of water moved by the sample, and removed the excess of water by pressing them against laboratory paper. We then weighed the samples again to determine their fresh weight and calculate water absorption capacity. Density of the moss sample was then estimated by dividing the dry weight by the volume of water moved by the samples in the pipette during the previous process. Further details on how we measured moss traits can be found in [25]. We also used growth forms (mats, turfs, and others) and whether mosses were pleurocarpous or acrocarpous as additional morphological traits.

Additionally, we measured six microscopic traits: cell length, width, form (length to width ratio), cell wall thickness, nerve type (i.e., null/short, long, or excurrent), and cell sculpture (i.e., smooth, mamillose/unipapillose, or pluripapillose). Cell measurements were mostly taken from the same samples used for leaf and entire moss traits. A few were obtained from new moss specimens collected from the same locations. We measured more than 60 medial cells from three leaves per species using an Olympus CH30 microscope. Images were then analysed with *ImageJ* to calculate their size. Finally, an average per species was calculated for further analyses. Species used in this study included: *Amblystegium serpens* (Amblystegiaceae)*, Anomodon viticulosus* (Thuidiaceae), *Brachythecium rivulare* (Brachytheciaceae)*, Bryum pseudotriquetrum* (Bryaceae)*, Cratoneuron filicinum* (Amblystegiaceae)*, Ctenidium molluscum* (Hypnaceae)*, Dialytrichia mucronata* (Pottiaceae)*, Didymodon sp* (Pottiaceae)*, Didymodon tophaceus* (Pottiaceae)*, Eucladium verticillatum* (Pottiaceae)*, Fissidens crassipes* (Fissidentaceae)*, Fissidens grandifrons* (Fissidentaceae)*, Fissidens rivularis* (Fissidentaceae)*, Fissidens taxifolius* (Fissidentaceae)*, Fontinalis antipyretica* (Fontinalaceae)*, Leptodictyum riparium* (Amblystegiaceae), *Oxyrrhynchium speciosum* (Brachytheciaceae)*, Palustriella commutata* (Amblystegiaceae)*, Philonotis caespitosa* (Bartramiaceae)*, Philonotis fontana* (Bartramiaceae)*, Plagiomnium undulatum* (Mniaceae)*, Pohlia melanodon* (Bryaceae)*, Rhizomnium punctatum* (Mniaceae)*, Rhynchostegiella teneriffae* (Brachytheciaceae)*, Rhynchostegium riparioides* (Brachytheciaceae)*, Scorpiurium circinatum* (Brachytheciaceae)*, Thamnobryum alopecurum* (Neckeraceae)*, Thuidium delicatulum* (Thuidiaceae), and *Trichostomum crispulum* (Pottiaceae).

### 2.2. Statistical Analyses

We first performed exploratory analyses to investigate the differences in elemental composition amongst the different moss species. To do so, we performed two cluster and two principal component (PCA) analyses. The first cluster and PCA analyses was performed with macro- and micronutrients (C, N, P, K, Ca, Na, Mg, and S), δ^13^C, δ^15^N, and the C:N, C:P, and N:P ratios. The second analyses were performed with trace elements, including Al, As, B, Cd, Co, Cr, Cu, Fe, Hg, Mn, Ni, Pb, and Zn. We defined different groups of species based on the separation of the different branches within the cluster analyses. We then overlaid these groups in a 2-dimensional representation of the PCA analyses (the two main axes extracted). The cluster analyses were based on Euclidean distances and the Ward D2 agglomeration method.

Secondly, we investigated the relationship between moss elemental composition and morphological and cell traits by performing sparse partial least squares (sPLS) regression analyses using the R package mixOmics [34]. sPLS is a multivariate analysis that allows us to investigate the relationship between two matrices (predictor and response matrices) by finding their underlying relationships through the extraction of latent variables. Additionally, the sparse method allows a selection of the predictor variables to facilitate the biological interpretation of the results. Three different sPLS models were fitted. The first model aimed to investigate how the elemental composition of mosses, δ^13^C, δ^15^N, and C:N, C:P, and N:P stoichiometric ratios were related to continuous moss cell traits (cell length, width, form, and cell wall thickness), with the latter being the response matrix. The second model was aimed at investigating the relationship between moss cell traits (cell length, width, form, cell wall thickness, nerve type, and cell sculpture) and all seven macroscopic morphological traits (leaf length, width, form and area, WAC, density, and MMA), with the latter being the response matrix. Nerve type and cell sculpture were coded as dummy variables to perform this model. The third model was aimed at investigating the relationship between all morphological traits and moss elemental concentrations, with the moss elemental concentration as the predictor matrix. We tuned the sPLS models to estimate the best choice in terms of number of components extracted per model and the number of predictors to be kept per component. We used three components for each final model and the number of predictors per component was estimated by the tuning of the sPLS. Results of the sPLS models were represented by using a clustered image map, and the variance explained from all response variables was assessed by linear models using the predicted values by all three components of the sPLS models.

Finally, we investigated whether different growth and life forms were related to specific elemental compositions. To do so, we performed two sparse partial least squares—discriminant analyses (sPLS-DA). sPLS-DA models are used for classification and discrimination of a categorical variable depending on multiple predictor variables, while allowing the selection of the most relevant predictors. We used the elemental composition of mosses as predictor variables, and acrocarpous/pleurocarpous as the response variable in one sPLS-DA model, with growth forms (mats, turfs, and others) as the response variable in the second model. Tuning of the model indicated that the best results were found when using three axes for the acrocarpous/pleurocarpous mosses, including 20, 13, and 20 predictors, respectively, on axes one, two and three. The tuning of the model for growth forms indicated that the best results were found when four axes were extracted, including 13, 20, 16, and 2 predictors, respectively. All statistical analyses were performed upon average values per species. All analyses were performed with R [35]. The code and data to perform these analyses is freely available at FigShare: https://doi.org/10.6084/m9.figshare.14916474.v1 (accessed on 6 July 2021)

## 3. Results

### 3.1. Elementome Similarity amongst Moss Species

Our analyses differentiated three main groups of mosses when clustering species based on their C:N:P stoichiometry, isotopic signatures of C and N (δ^13^C and δ^15^N), and macronutrients (C, N, P, K, Na, Mg, Ca, and S) (Appendix A). The first axis of the PCA was mainly negatively related to the concentration of N, P, and S and positively to C:N, C:P, and N:P ratios. The second axis was mainly negatively related to the concentration of C, and positively related to Mg, K, Na, and δ^13^C. The grey group, containing species such as *F. crassipes* and *T. crispulum*, was mainly characterised by having low concentrations of N, P, K, Mg, and Na, and high C, C:N, C:P, and N:P ratios. Conversely, the orange group, containing species such as *A. serpens* and *S. circinatum*, had particularly high concentrations of N, P, and S and low C:N, C:P, and N:P ratios. Similarly, the blue group, including only *F. antypiretica* and *L. riparium*, was characterised for having high C, N, P, and S, and very low C:N, C:P, and N:P, Na, K, and Mg.

We identified four main groups when clustering species using their concentration of trace elements and heavy metals (Al, As, B, Cd, Co, Cr, Cu, Fe, Hg, Mn, Pb, Ni, and Zn) (Appendix A). The first axis of the PCA was clearly negatively related to the concentration of most of the elements, while the second axis was positively related to the concentration of B and Fe and negatively related to Pb and Cd. The orange group, including species such as *B. rivulare* and *F. taxifolius*, was characterised for having the highest concentrations of trace elements. The black group, occupying the central region of the two-dimensional space, and including species such as *C. molluscum* and *B. pseudotriquetrum*, presented intermediate concentrations of trace elements. Finally, the blue and yellow groups were placed at low values of trace elements, but while the blue group (e.g., *Didymodon* sp. and *D. tophaceus*) presented high values of B and low values of Pb and Cd, the yellow group (e.g., *F. antypiretica* and *F. rivularis*) presented the opposite behaviour.

### 3.2. Cell Traits and Their Relationship with Moss Elemental Composition and Macroscopic Traits

We found statistically significant differences in cell length, width, form (length to width ratio), and cell wall thickness amongst species (Figure 1). Our measurements were very similar to those from previous authors for the same species [36,37]. Our statistical models indicated that moss species with higher concentrations of P and Zn had longer and narrower cells compared to those presenting lower P and Zn concentrations (Figure 2). Conversely, C and δ^13^C were negatively related to cell length and form. Variability in cell width and cell wall thickness, however, was not well explained by the elemental composition of mosses.

On the other hand, cell traits, overall, were good predictors of moss macroscopic traits (Figure 3). Mosses with wider and longer cells, long nerves, and thicker cell walls had generally wider, longer, and bigger leaves with a low length to width ratio. Moss mass per area was strongly negatively linked to smooth, long, and narrow cells, while pluripapillose cells were positively correlated with high moss mass per area. The capacity to absorb water of mosses was lower in those species with narrower pluripapillose cells and thicker cell walls. Moss density, however, was not well explained by moss cell traits.

### 3.3. Relationship between Moss Elemental Composition, Morphological Traits, and Growth Forms

Our analyses showed that moss elemental composition could explain variability in morphological traits reasonably well for six out of seven traits: leaf length, width, area and form, MMA, and density (Figure 4). Water absorption capacity (WAC) was not well explained by the elemental composition of mosses. Mosses with big leaves (i.e., large area, long, and wide) were more likely to present higher concentrations of C and low concentrations of metals (e.g., Cu, Cr, Ni, Fe…) and micronutrients (e.g., Na, Mg, Ca, and B) than mosses with smaller leaves. On the other hand, high C:N, N:P, and C:P ratios were related to high MMA, density, and needle-like leaves. These correlations with C:N:P stoichiometry, however, emerged due to a negative relationship between N and P with density, MMA, and leaf form, and not due to a strong positive correlation with C.

Growth forms were also reasonably well explained by the elemental composition of mosses (Figure 5). The sPLS-DA model for acrocarpous vs. pleurocarpous mosses correctly classified 96.6% (28 out of 29 species, using two components) of the species based on their elemental composition. The model predicting mats, turfs, and other growth forms successfully classified 86.2% (25 out of 29, four components) of the moss species based on their elementomes. Both models showed very similar patterns in their two first axes, although the second axes were inverted. In both models, the first axes were positively related to trace metals and micronutrients, and negatively related to C:N. The second axis in the acrocarpous/pleurocarpous model was negatively related to B and positively related to several heavy metals such as Cd, Pb, Co, and Zn, while the opposite was shown in the model predicting growth forms. Hence, acrocarpous and turf-forming mosses were more likely to present low concentrations of most macro- and micronutrients, and high C:N, C:P, and N:P ratios. Conversely, pleurocarpous and mat-forming mosses presented higher concentrations of macro- and micronutrients and lower C:N, C:P, and N:P ratios.

## 4. Discussion

### 4.1. Elementome Differences between Species

Our exploratory analyses indicated that there were clear differences in the elementomes of the different species, enabling the classification of different groups of species depending on their elementome similarity (Appendix A). However, the grouping of species differed depending on whether macronutrients or micronutrients and trace elements were used, suggesting a weak or non-existent relationship between both groups of elements across species. This result further indicates that each element may provide unique information regarding the functioning of the organisms and, therefore, studies using only macronutrients such as C, N, and P may be missing an important set of features of the organisms under study, as suggested by several authors [18,38].

On the other hand, our results indicated that morphologically similar species, and those from the same genus (e.g., *Fissidens*) were often included in different groups, both using macro- and micronutrients and trace elements. This result suggests that elementome plasticity may, under certain circumstances, be larger than that of morphological traits. However, the strong relationship found between moss elemental concentration and micro- and macroscopic morphological traits (Figure 2 and Figure 4) clearly denotes that particular elementomes are related to particular traits. Additionally, different elemental compositions are indicative of metabolic differences [39], and that could explain why, despite their strong morphological similarity, different *Fissidens* species are found in different habitats. Experiments aimed to analyse the intraspecific variability in moss traits and elementomes will be needed to investigate whether the intraspecific plasticity in morphological traits is larger than that of elementomes or vice versa.

### 4.2. Moss Elemental Composition Controls Micro- and Macroscopic Morphological Traits across Species

Our results clearly confirmed our initial hypothesis stating that bryophyte elementomes are related to their micro- and macroscopic morphological traits. In fact, we found that moss elementomes can explain a large proportion of the variability of most morphological traits and growth forms across species (Figure 2 and Figure 4). Hence, our results fully support previous studies, based on vascular plants, indicating that the elemental composition of organisms plays a paramount role in determining their functional traits [15,22]. Nonetheless, our results do not disregard the direct or indirect effect of abiotic factors on either the elemental composition of bryophytes or their morphological traits. Previous research has shown that water chemistry has an important impact on bryophyte elemental concentrations [9,30], and their functional and morphological traits [25,31]. Additionally, the evolutionary history of bryophytes has also been shown to be a relevant predictor of their elemental concentration and their functional traits [25,30]. However, it is very difficult to identify the primary force driving the observed correlation between bryophyte elemental concentration and morphological traits due to a two-way interaction between traits and elemental concentrations: some bryophyte traits may facilitate or hinder the absorption of certain elements, but at the same time, certain amounts of determined elements are required to build particular plant structures. Debate on that matter, therefore, deserves further attention.

We found that moss traits related to mosses living in springs with hard water (i.e., needle-like leaves, high density and MMA [25]) had higher C:N, C:P, and N:P ratios, indicating a possible N and P limitation that could, consequently, hinder their growth and competitive potential (e.g., *E. verticillatum*, *D. tophaceus, T. crispulum,*
Appendix A). Nutrient-poor species, however, may follow a conservative strategy that allows them to survive under stressful conditions that other, more productive species, may not tolerate [15]. Nonetheless, several species presenting big and wide leaves and low density such as *F. antipyretica*, *L. riparium* or *R. riparioides* presented low C:N, C:P, and N:P ratios. The higher proportion of N and P in relation to C clearly allows these species to grow faster and dominate their environments better, but they may fail to survive under stressful environments. These findings agree with previous research based on vascular plants [22,40] and partially support our second hypothesis stating that nutrient-rich mosses would generally be less dense, have lower MMA and a low length-to-width ratio for leaves. However, statistical evidence indicating that high N and P concentrations correlate with larger leaves was not clearly found. Additionally, these results and interpretations perfectly agree with previous research performed with bryophytes, indicating that high N and P are related to high photosynthetic capacity and low moss mass per area [23,24]. However, although the elemental composition of mosses was a very good predictor of acrocarpous and pleurocarpous growth forms, there was no clear relationship regarding whether higher C:N, C:P, and N:P ratios were linked to any particular category, as trace elements, such as Cd, Pb, and B, also played an important role predicting them. Combined with the important effect of micronutrients and trace elements on morphological traits, this result further supports our abovementioned statement suggesting that a larger number of elements (i.e., multidimensional information) is needed in order to capture the morphological and, potentially, functional variability between organisms when comparing elementomes.

The relationship we found between trace elements and morphological traits is, hence, completely new in the field, and represents the first attempt to investigate their relationship in bryophytes. We found that different micronutrients and trace elements were related to morphological traits in different ways. While Pb, Cd, Cr, and Cu presented negative relationships with leaf form, density, and MMA, most micronutrients and trace elements, such as Al, As, Ni, Cr, Cu, Fe, Mg, or Mn, were clearly negatively related to leaf length and area. Therefore, we hypothesise that certain morphological traits of bryophytes may impede or facilitate the absorption and bioaccumulation of trace elements, similar to that which has been suggested for cell traits [41] such as cell wall thickness [42]. According to our results, mosses with wide and big leaves, and low MMA would be more likely to accumulate elements such as Pb or Cd, while those with shorter and narrower leaves could be better accumulators of Al, As, Ni, Cr, Cu, or Mn. These results could help researchers to improve their biomonitoring results by selecting species with the morphological traits that are better suited for monitoring the elements of interest, should our hypothesis hold true after future research.

On the other hand, the elemental composition of mosses was not as strongly related to cell traits as to macroscopic traits (Figure 2), and only four elements appeared as important predictors of cell traits. These results suggest that differences in the elemental composition amongst species may be more important for whole-organism processes of moss functioning, fitness, and adaptation or evolution (e.g., overall growth rate, photosynthesis, drought tolerance, reproduction [19,31]) than for processes occurring at the cellular level. Nonetheless, cell traits were good predictors of moss macroscopic morphological traits, which actually indicates a good correspondence between cell and macroscopic traits (Figure 3). Considering additional cell traits could potentially help in solving these contrasting results.

Overall, we here present the first results relating moss micro- and macroscopic morphological traits and growth forms with their elemental composition, showing a clear link between them. Further studies focused on the intraspecific variability on moss elemental composition and morphological traits will be interesting in order to investigate whether the link between elementomes and morphological traits occur only amongst species, or if it also occurs within species. Given that mosses conserve functional and morphological traits present in the early plants, our results represent an important step towards understanding how the elemental composition may have influenced morphological and, potentially, functional traits throughout the evolution and colonisation of land plants.

## Figures and Tables

**Figure 1 plants-10-01581-f001:**
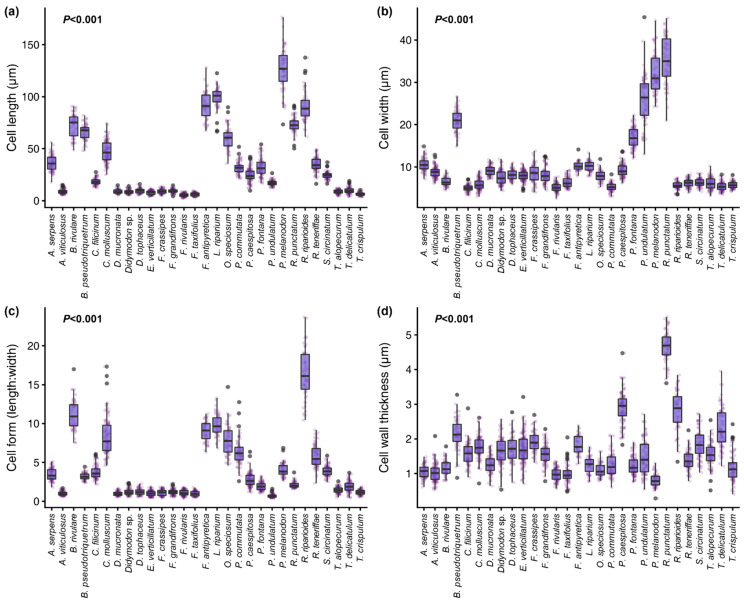
Boxplots showing measurements of cell traits per species. All measured traits presented statistically significant differences amongst species at the <0.001 level (ANOVA test).

**Figure 2 plants-10-01581-f002:**
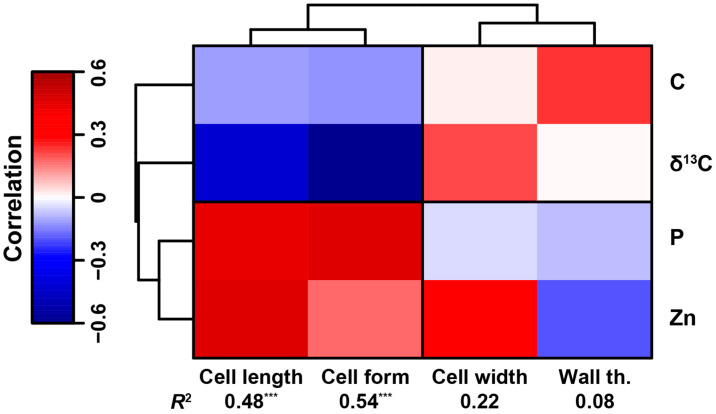
Clustered heat map showing the relationship between concentration of elements (**right**) and cell traits (**bottom**). The *R*^2^ shown below cell traits correspond to the variance explained by a linear combination of axes 1, 2, and 3 of the sPLS model. Significance levels: *** *p* < 0.001. See Methods for further information on the models. Acronyms: Wall th.: wall thickness.

**Figure 3 plants-10-01581-f003:**
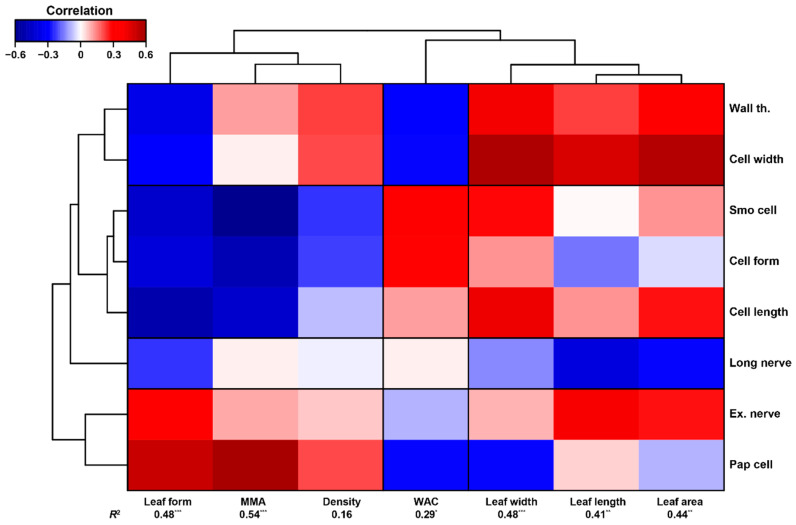
Clustered heat map showing the relationship between cellular (**right**) and macroscopic traits (**bottom**). The *R*^2^ shown below morphological traits correspond to the variance explained by a linear combination of axes 1, 2, and 3 of the sPLS model. Significance levels: * *p* < 0.05, ** *p* < 0.01, *** *p* < 0.001. See Methods for further information on the models. Acronyms: Wall th.: wall thickness, Smo cell: smooth cell, Ex. Nerve: excurrent nerve, and Pap cell: pluripapillose cell.

**Figure 4 plants-10-01581-f004:**
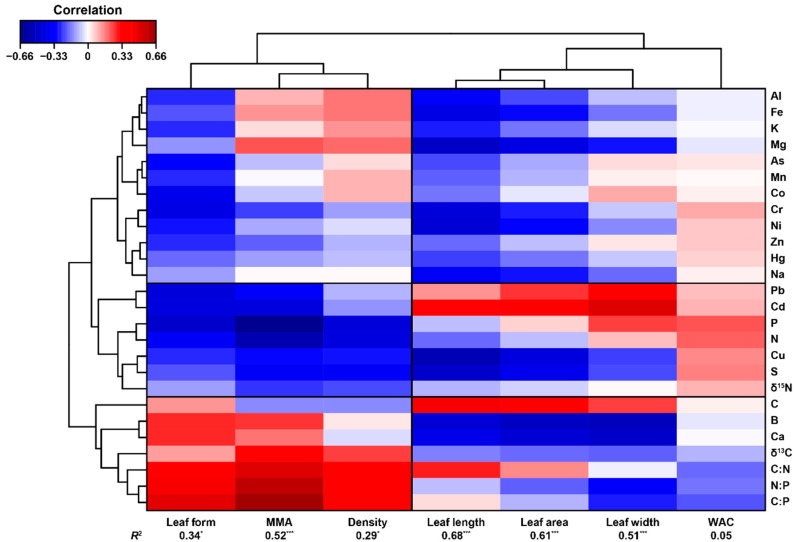
Clustered heat map showing the relationship between concentration of elements (**right**) and macroscopic morphological traits (**bottom**). The *R*^2^ shown below morphological traits correspond to the variance explained by a linear combination of axes 1, 2, and 3 of the sPLS model. Significance levels: * *p* < 0.05, *** *p* < 0.001. See Methods for further information on the models.

**Figure 5 plants-10-01581-f005:**
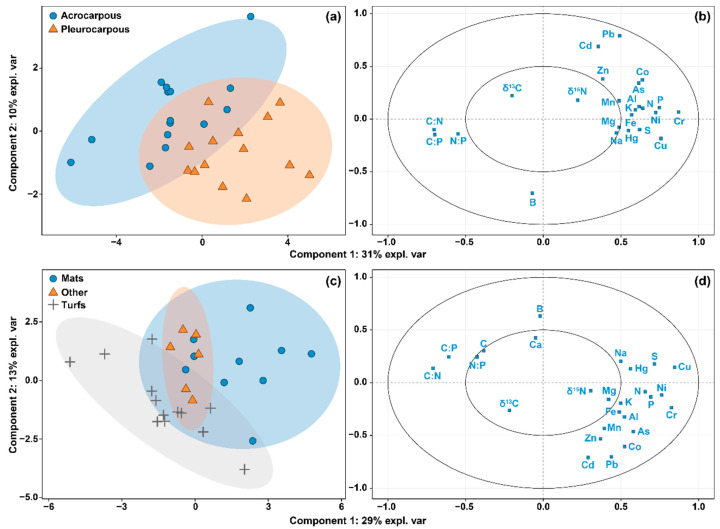
Graph showing the scores (panels (**a**,**c**)) and variable correlations (**b**,**d**) in a two-dimensional space of two sPLS-DA models classifying the growth form of 29 moss species by means of their elemental composition. Panels (**a**,**b**) show the model for acrocarpous vs. pleurocarpous types, and panels (**c**,**d**) show the model for mats, turfs, and other growth forms. Models classified acrocarpous and acrocarpous species with an accuracy of 93.1% (28 out of 29 species, using two components) and mats, turfs, and other growth forms with an accuracy of 86.2% (25 out of 29, four components). Shaded areas represent the 95% confidence interval of each group.

## Data Availability

All data and code used in this study is freely available at Figshare: https://doi.org/10.6084/m9.figshare.14916474.v1 (accessed on 7 July 2021).

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
