# Peer review of "Do Bryophyte Elemental Concentrations Explain Their Morphological Traits?"

_plants, 2021, doi:10.3390/plants10081581_

Round 1
Reviewer 1 Report
Review of Fernández-Martínez et al.: Do bryophyte elemental concentrations explain their morphological traits?
There are so many parameters influencing the growth forms (incl. cell length, etc.) of bryophytes. There were no measures regarding microclimatic parameters, light, etc. in this study. There were even no investigations of water chemistry or chemical composition of the sampling areas. However, all these factors have an even stronger influence on the growth traits than elemental concentrations. Maybe elemental concentrations reflect some of these parameters. But by missing data on these factors, this cannot be confirmed.
Furthermore, the species in quest were obviously from different sites with differing environmental conditions and differing geology, and probably differing water composition. At least there is nothing given in the text regarding the sites where the samples come from. Therefore, the results on the investigated parameters could be coincidental.
Many of these tested species are not aquatic species; maybe these grow in the surrounding springs, but certainly, many of these are not growing all year round in the water, as stated. Therefore the species are hardly comparable.
There seem to be some parallels to previously published papers. What is new compared to previous published studies by the authors, especially [25]?
Minor revisions:
Line 30: I do not agree that bryophytes are simple at all – maybe regarding their morphological features (which is not true at all…), however their physiology is even more complex than those of vascular plants.
Line 33: Bryophytes accumulate substances under all environmental conditions, not even under environmental pollution.
Line 43: “functional traits” should be explained – which traits do you mean? There are many and just a few which are influenced or investigated by the study at hand. The authors just mention “growth form” in this place as one of many traits – refer also to microscopic, macroscopic, physiological etc.
Line 85 ff: was there an analysis of the elements in spring water?
Line 96 ff: how were the samples digested?
Line 130 ff: many of these species are not aquatic species; maybe these grow in the surrounding of springs, but certainly many of these are not growing all year round in water, as stated above (line 85)
There are no data on water chemistry, environmental microclimatic parameters, light intensity etc.
Statistics seem to be ok
Fig. captures: Why are these above the Figure? They should be placed below
Author Response
Reviewer 1:
Q1: There are so many parameters influencing the growth forms (incl. cell length, etc.) of bryophytes. There were no measures regarding microclimatic parameters, light, etc. in this study. There were even no investigations of water chemistry or chemical composition of the sampling areas. However, all these factors have an even stronger influence on the growth traits than elemental concentrations. Maybe elemental concentrations reflect some of these parameters. But by missing data on these factors, this cannot be confirmed.
R1: We agree with the referee that abiotic factors may influence morphological and functional traits of bryophytes and we acknowledge that in the introduction, see L. 55-59: “Recent publications focused on bryophyte functional traits have considerably increased our knowledge regarding the relationship between the environment and bryophyte functional traits [25–29]. In particular, some of these studies identified water chemistry (e.g., pH, dissolved nutrients and heavy metals) as an important determinant of moss traits and growth forms [2,25,28].” However, the aim of the present study is to test whether moss chemical composition relates to their functional traits, independently of their environment and how the environment can affect their internal chemistry, which is actually the missing information in this regard, see L. 65-70: “However, to the best of our knowledge, no efforts have been devoted to investigate whether the elemental composition of bryophytes is related to their micro- and macroscopic morphological traits and growth forms across a large number of species. To fill in this gap, we here analysed 80 samples from 29 aquatic and semi-aquatic (hygrophytic) moss species to investigate the relationship between gametophyte moss nutrient concentrations and their morphological traits”. We hope the reviewer and the editor can agree that our line of research is equally interesting as those studying how the environment affects bryophyte traits (see comments by reviewer 2).
Q2: Furthermore, the species in quest were obviously from different sites with differing environmental conditions and differing geology, and probably differing water composition. At least there is nothing given in the text regarding the sites where the samples come from. Therefore, the results on the investigated parameters could be coincidental.
R2: We thank the referee for this comment and acknowledge that this information was missing in the previous version of the manuscript. Following the reviewer’s advice (and reviewer’s 3 comment), we have now provided more information on the chemical composition of the sampled springs and their chemical composition, see L. 87-90: “In the subset of springs used here, water pH ranged from 5.17 to 8.34 (median: 7.20) and water conductivity ranged from 24 to 2094 (median: 498) µS cm-1. A detailed analysis of the water chemical composition can be found in the online material from references [25,30,31]”. Please, notice that the data on the water chemical composition of our springs are publically available at Figshare (references 25, 30 and 31).
Q3: Many of these tested species are not aquatic species; maybe these grow in the surrounding springs, but certainly, many of these are not growing all year round in the water, as stated. Therefore, the species are hardly comparable.
R3: The reviewer is correct: some of the tested species are not aquatic. However, they are all hygrophytic, this is that they live in sites of high humidity. In our case, all species and samples were in continuous (or nearly continuous) contact with water (e.g., submerged, or receiving drops from the spring) as stated in the Methods section of our manuscript. Nonetheless, and even if species were collected from different habitats, we cannot see any reason why species would not be comparable for the question under study. Consider all the publications regarding the leaf economic spectrum (e.g.,. (Wright et al. 2004)), in which they compare species from all over the world. Those species live in completely different habitats and yet patterns relating their elemental composition with their functional traits emerge. Therefore, we see no reason for concern regarding the comparability of our species.
References:
Wright, I.J., Reich, P.B., Westoby, M., Ackerly, D.D., Baruch, Z., Bongers, F., et al. (2004). The worldwide leaf economics spectrum. Nature, 428, 821–7.
Q4: There seem to be some parallels to previously published papers. What is new compared to previous published studies by the authors, especially [25]?
R4: Yes, there are parallels, as this paper is intended to follow a common line of research. However, the differences between this study and reference 25 (Fernández‐Martínez et al. 2019) is very clear: while in ref 25 we were investigating how abiotic (climate, water chemistry) and phylogenetic factors affect moss traits, in the present study we are investigating how the elemental composition of mosses are related to their morphological traits, independently of their environment. As shown in another of our studies, the chemical composition of mosses is highly species-specific (Fernández‐Martínez et al. 2021), for which, despite the effect of the environment on moss elemental composition, moss traits can relate to their elemental composition independently.
References:
Fernández‐Martínez, M., Berloso, F., Corbera, J., Garcia‐Porta, J., Sayol, F., Preece, C., et al. (2019). Towards a moss sclerophylly continuum: Evolutionary history, water chemistry and climate control traits of hygrophytic mosses. Funct. Ecol., 33, 2273–2289.
Fernández‐Martínez, M., Preece, C., Corbera, J., Cano, O., Garcia‐Porta, J., Sardans, J., et al. (2021). Bryophyte C:N:P stoichiometry, biogeochemical niches and elementome plasticity driven by environment and coexistence. Ecol. Lett., 24, 1375–1386.
Minor revisions:
Q5: Line 30: I do not agree that bryophytes are simple at all – maybe regarding their morphological features (which is not true at all…), however their physiology is even more complex than those of vascular plants.
R5: Following the reviewer’s comment, we have removed that statement regarding the simplicity of bryophytes. The text now reads (L.30-31): “Bryophytes are amongst the most fascinating organisms of the plant kingdom, given their ecology, physiology and morphology”.
Q6: Line 33: Bryophytes accumulate substances under all environmental conditions, not even under environmental pollution.
R6: Following the reviewer’s comment, we have now rephrased that sentence, reading (L. 33-36): “Particularly under environmental pollution, bryophytes have been shown to accumulate heavy metals and change their elemental composition or, in the event that they cannot tolerate the new conditions, they disappear”. We now highlight that this has been shown particularly under environmental pollution.
Q7: Line 43: “functional traits” should be explained – which traits do you mean? There are many and just a few which are influenced or investigated by the study at hand. The authors just mention “growth form” in this place as one of many traits – refer also to microscopic, macroscopic, physiological etc.
R7: Following the reviewer’s comment we have clarified our sentence in old L. 43, now reading (L. 43-45): “The elemental composition of organisms has been repeatedly shown to be a very good indicator of their morphological and functional traits (e.g., photosynthesis, respiration), the ecological strategies they follow, and their relationship with their environment”. Please, notice that in the line suggested by the reviewer there is no mention of growth form, but it is just a general introduction to the relationship between functional traits and the elemental composition of organisms.
Q8: Line 85 ff: was there an analysis of the elements in spring water?
R8: Yes, there were, but these data are not included in this paper because it lies out of its scope. We have, nonetheless, included a brief summary of the water pH and electric conductivity, see L. 87-90: “In the subset of springs used here, water pH ranged from 5.17 to 8.34 (median: 7.20) and water conductivity ranged from 24 to 2094 (median: 498) µS cm-1. A detailed analysis of the water chemical composition can be found in the online material from references [25,30,31]”. Please, notice that the data on the water chemical composition of our springs are publically available at Figshare (references 25, 30 and 31).
Q9: Line 96 ff: how were the samples digested?
R9: IRMS does not require a previous digestion of the samples. We digested the samples for ICP-MS analysis, described in L. 106-107 “[…] after the samples were digested overnight at 90º C with nitric acid and hydrogen peroxide in a 2:1 ratio”.
Q10: Line 130 ff: many of these species are not aquatic species; maybe these grow in the surrounding of springs, but certainly many of these are not growing all year round in water, as stated above (line 85)
R10: All these species are hygrophytic and we can assure the referee that they were all collected from springs, and they were receiving water from the spring when collected.
Q11: There are no data on water chemistry, environmental microclimatic parameters, light intensity etc.
R11: We have these data, but as stated above, that research question was investigated in our previous publication Fernández-Martínez et al., 2019 – Functional Ecology.
Q12: Statistics seem to be ok
R12: Thanks for the comment.
Q13: Fig. captures: Why are these above the Figure? They should be placed below
R13: We are now following the template provided by the journal. Some figure captures have been moved below the figures, but some other have been left on top, presumably like the journal requests them.
Reviewer 2 Report
Over the last 150 years, much information has been collected how environmental conditions influence plant morphology. However, plants inhabiting the very same habitat and even the very same ecological niche may exhibit a very different morphology, though they are exposed to the same environmental conditions. The reasons for these differences are only poorly understood. In this study, the authors test the hypothesis that elemental composition correlates with morphology, and present evidence that this is indeed the case, at least in bryophytes from a very specific type of habitat. This hypothesis is fascinating, and will hopefully stimulate a multitude of future studies. After all, this study only states the fact that there is a correlation, but offers neither a mechanism, how this could work, nor allows it to decide if element composition controls morphology, or vice versa, or if some external factor (water chemistry?) controls both. However, answering these questions would be far beyond the scope of a single publication, and within this scope, the authors present thoroughly conducted research. Thus, I have only minor comments:
It is certainly relevant that the authors assessed the influence of the systematic position of their test species, as done by the itemisation as „acrocarpous“ and „pleurocarpous“ in figure 5, and there is no need to modify this figure. However, this is only a very course classification. Wouldn’t it make sense to check, if there are differences between orders and families? Furthermore, I feel the authors should address an alternative hypothesis: mosses absorb elements via there whole surface, and have very limited capabilities to keep anything outsider their organisms. Thus, maybe it is not the element composition which predicts the morphology of a moss, but the chemistry of the surrounding water predicts both element composition and morphology? Hereby, I do not suggest that any additional measurements should be done (though this might be an interesting topic for a follow-up study), but the authors should discuss this possibility. Furthermore, it might be interesting to consider also the number and size of chloroplasts in future studies!
Finally, two very minor issues:
L 105: Maybe, the measurement of „moss density“ could be defined in more detail?
L 130: The families should be added.
Author Response
Reviewer 2:
Q1: Over the last 150 years, much information has been collected how environmental conditions influence plant morphology. However, plants inhabiting the very same habitat and even the very same ecological niche may exhibit a very different morphology, though they are exposed to the same environmental conditions. The reasons for these differences are only poorly understood. In this study, the authors test the hypothesis that elemental composition correlates with morphology, and present evidence that this is indeed the case, at least in bryophytes from a very specific type of habitat. This hypothesis is fascinating, and will hopefully stimulate a multitude of future studies. After all, this study only states the fact that there is a correlation, but offers neither a mechanism, how this could work, nor allows it to decide if element composition controls morphology, or vice versa, or if some external factor (water chemistry?) controls both. However, answering these questions would be far beyond the scope of a single publication, and within this scope, the authors present thoroughly conducted research. Thus, I have only minor comments:
R1: We are very happy to read such a positive assessment of our work. Thanks for your words, they are really appreciated.
Q2: It is certainly relevant that the authors assessed the influence of the systematic position of their test species, as done by the itemisation as „acrocarpous“ and „pleurocarpous“ in figure 5, and there is no need to modify this figure. However, this is only a very course classification. Wouldn’t it make sense to check, if there are differences between orders and families? Furthermore, I feel the authors should address an alternative hypothesis: mosses absorb elements via there whole surface, and have very limited capabilities to keep anything outsider their organisms. Thus, maybe it is not the element composition which predicts the morphology of a moss, but the chemistry of the surrounding water predicts both element composition and morphology? Hereby, I do not suggest that any additional measurements should be done (though this might be an interesting topic for a follow-up study), but the authors should discuss this possibility. Furthermore, it might be interesting to consider also the number and size of chloroplasts in future studies!
R2: Thanks again for this constructive comment. Following the reviewer’s comment, we have now added some discussion regarding the effect of water chemistry on bryophyte elemental composition and morphological traits, see L. 322-333: “Nonetheless, our results do not disregard the direct or indirect effect of abiotic factors on either the elemental composition of bryophytes or their morphological traits. Previous research has shown that water chemistry has an important impact on bryophyte elemental concentrations [9,30], and their functional and morphological traits [25,31]. Additionally, the evolutionary history of bryophytes has also been shown to be a relevant predictor of their elemental concentration and their functional traits [25,30]. However, it is very difficult to identify the primary force driving the observed correlation between bryophyte elemental concentration and morphological traits because of a two-way interaction between traits and elemental concentrations: some bryophyte traits may facilitate or hinder the absorption of certain elements, but at the same time, certain amounts of determined elements are required to build particular plant structures. Debate on that matter, therefore, deserves further attention”.
Regarding differences between orders and families, we tested the phylogenetic relationship on moss morphological traits and elemental composition in two of our previous publications (Fernández-Martínez et al., 2019; Fernández-Martínez et al., 2021), this is the reason why we did not include that information in the current manuscript (to avoid repeating results already published). We have also added this information in the current version of the manuscript (see text above).
References:
Fernández‐Martínez, M., Berloso, F., Corbera, J., Garcia‐Porta, J., Sayol, F., Preece, C., et al. (2019). Towards a moss sclerophylly continuum: Evolutionary history, water chemistry and climate control traits of hygrophytic mosses. Funct. Ecol., 33, 2273–2289.
Fernández‐Martínez, M., Preece, C., Corbera, J., Cano, O., Garcia‐Porta, J., Sardans, J., et al. (2021). Bryophyte C:N:P stoichiometry, biogeochemical niches and elementome plasticity driven by environment and coexistence. Ecol. Lett., 24, 1375–1386.
Q3: Finally, two very minor issues:
L 105: Maybe, the measurement of „moss density“ could be defined in more detail?
R3: The description of moss density can be found in L. 118-125: “Water absorption capacity was determined by calculating the fresh-to-dry weight ratio of the same moss sample used to calculate MMA. We rehydrated the samples by submerging them into distilled water for three minutes in a graduated cylinder of 10 ml. We then recorded the volume of water moved by the sample and removed the excess of water by pressing them against laboratory paper. We then weighed the samples again to determine their fresh weight and calculate water absorption capacity. Density of the moss sample was then estimated by dividing the dry weight by the volume of water moved by the samples in the pipette during the previous process”. We have also added “(moss dry mass per volume)” in L 111-112 to clarify the concept, as requested.
Q4: L 130: The families should be added.
R4: Following the reviewer’s comment, we have now added all family names, see L. 137-149: “Species used in this study included Amblystegium serpens (Amblystegiaceae), Anomodon viticulosus (Thuidiaceae), Brachythecium rivulare (Brachytheciaceae), Bryum pseudotriquetrum (Bryaceae), Cratoneuron filicinum (Amblystegiaceae), Ctenidium molluscum (Hypnaceae), Dialytrichia mucronata (Pottiaceae), Didymodon sp (Pottiaceae), Didymodon tophaceus (Pottiaceae), Eucladium verticillatum (Pottiaceae), Fissidens crassipes (Fissidentaceae), Fissidens grandifrons (Fissidentaceae), Fissidens rivularis (Fissidentaceae), Fissidens taxifolius (Fissidentaceae), Fontinalis antipyretica (Fontinalaceae), Leptodictyum riparium (Amblystegiaceae), Oxyrrhynchium speciosum (Brachytheciaceae), Palustriella commutata (Amblystegiaceae), Philonotis caespitosa (Bartramiaceae), Philonotis fontana (Bartramiaceae), Plagiomnium undulatum (Mniaceae), Pohlia melanodon (Bryaceae), Rhizomnium punctatum (Mniaceae), Rhynchostegiella teneriffae (Brachytheciaceae), Rhynchostegium riparioides (Brachytheciaceae), Scorpiurium circinatum (Brachytheciaceae), Thamnobryum alopecurum (Neckeraceae), Thuidium delicatulum (Thuidiaceae) and Trichostomum crispulum (Pottiaceae)”.
Reviewer 3 Report
The main question that arises in me reading this manuscript is, as the title states, if elemental concentrations can really explain their morphological traits. This is the idea that is conveyed throughout the manuscript, from the title to the last lines (367-368): “…understanding how the elemental composition may have influenced morphological and, potentially, functional traits”. But, could it not be the opposite, the presence of certain morphological traits explain the elemental concentrations? In fact, towards the end of the manuscript (line 341 and following) it is pointed out that “certain morphological traits of bryophytes may impede or facilitate the absorption and bioaccumulation of trace elements”. I think this should be clarified. That is, which would be the independent variable here and which would be the dependent variable?
Lines 40-41. The authors note the lack of data on concentrations of elements in bryophytes from uncontaminated areas. The data in this work comes from supposedly clean sites, which I find interesting. However, from what I can deduct from the map, there are points very close to the city of Barcelona and there are elements, such as Hg, that are dispersed very efficiently by air. Perhaps the data could be compared with that of other publications that present data on natural concentrations or background levels in bryophytes, to assess whether these concentrations are typical of clean sites. Or at least present a table in the paper itself (there is one online) with mean, median, maximum and minimum concentrations, so that readers can get an idea of the levels in the samples.
Lines 68-70. It is indicated that the elemental composition in vascular plants has been used as an indicator of their morphological traits. The authors could include some references about this.
Lines 84-85. Although an article is cited in relation to the chemical characteristics of the water where the mosses grew, it would be interesting if the authors provide some data of these characteristics. Perhaps average values of pH and conductivity, with maximum and minimum of the total sampling points.
Line 92. How long were the samples immersed in acetic acid?
Figure 2. I would add x-axis data labels to the top two graphs. It is difficult to know which species each boxplot corresponds to.
Author Response
Reviewer 3:
Q1: The main question that arises in me reading this manuscript is, as the title states, if elemental concentrations can really explain their morphological traits. This is the idea that is conveyed throughout the manuscript, from the title to the last lines (367-368): “…understanding how the elemental composition may have influenced morphological and, potentially, functional traits”. But, could it not be the opposite, the presence of certain morphological traits explain the elemental concentrations? In fact, towards the end of the manuscript (line 341 and following) it is pointed out that “certain morphological traits of bryophytes may impede or facilitate the absorption and bioaccumulation of trace elements”. I think this should be clarified. That is, which would be the independent variable here and which would be the dependent variable?
R1: We thank the reviewer for this interesting comment. To actually solve this question, we should probably perform experiments, but still the question would be very difficult to answer, like the reviewer points out. From our perspective, we are more inclined to think that, initially, the elemental composition of organisms is what determines their morphology (e.g., stinging hairs of stinging nettles are Si-enriched structures, and therefore they need Si to perform their function but they are not able to absorb it; other plant structures do). However, we fully agree with the referee that certain plant functional traits facilitate the incorporation of elements within their organisms and, therefore, functional traits should determine those elemental compositions. Nonetheless, the aim of this study, as pointed out by the reviewer, was to tell whether just by knowing the elemental composition of organisms we could infer their morphological traits, no matter what the origin of this correlation is (see also comments by referee 2). Therefore, and following the reviewer’s suggestion, we now elaborate and clarify this issue throughout the manuscript, see L. 322-333: “Nonetheless, our results do not disregard the direct or indirect effect of abiotic factors on either the elemental composition of bryophytes or their morphological traits. Previous research has shown that water chemistry has an important impact on bryophyte elemental concentrations [9,30], and their functional and morphological traits [25,31]. Additionally, the evolutionary history of bryophytes has also been shown to be a relevant predictor of their elemental concentration and their functional traits [25,30]. However, it is very difficult to identify the primary force driving the observed correlation between bryophyte elemental concentration and morphological traits because of a two-way interaction between traits and elemental concentrations: some bryophyte traits may facilitate or hinder the absorption of certain elements, but at the same time, certain amounts of determined elements are required to build particular plant structures. Debate on that matter, therefore, deserves further attention”.
Q2: Lines 40-41. The authors note the lack of data on concentrations of elements in bryophytes from uncontaminated areas. The data in this work comes from supposedly clean sites, which I find interesting. However, from what I can deduct from the map, there are points very close to the city of Barcelona and there are elements, such as Hg, that are dispersed very efficiently by air. Perhaps the data could be compared with that of other publications that present data on natural concentrations or background levels in bryophytes, to assess whether these concentrations are typical of clean sites. Or at least present a table in the paper itself (there is one online) with mean, median, maximum and minimum concentrations, so that readers can get an idea of the levels in the samples.
R2: We understand the point of the reviewer regarding presenting these values in the manuscript. However, please, notice that these data are already publicly available at Figshare (two entries, one for this paper and another one for our paper Fernández-Martínez et al., 2021 Ecology Letters) and even presented in another study in a table in the Supplementary Material (Fernández-Martínez et al., 2021 - Freshwater Biology). All these papers are referenced in the manuscript, so we think we should not include these data as a table in the current manuscript to avoid self-plagiarism (also including only a subset of the data we collected for the other studies). We hope the reviewer and the editor can agree with our decision.
Q3: Lines 68-70. It is indicated that the elemental composition in vascular plants has been used as an indicator of their morphological traits. The authors could include some references about this.
R3: Following the reviewer’s suggestion, we have now included three references to support that claim, see L. 69-71: “We hypothesised that the elemental composition of moss species will be a very good indicator of their morphological traits, similar to that which has been reported in vascular plants (Wright et al. 2004; Díaz et al. 2016; Peñuelas et al. 2019)”.
References:
Díaz, S., Kattge, J., Cornelissen, J.H.C., Wright, I.J., Lavorel, S., Dray, S., et al. (2016). The global spectrum of plant form and function. Nature, 529, 167–171.
Peñuelas, J., Fernández‐Martínez, M., Ciais, P., Jou, D., Piao, S., Obersteiner, M., et al. (2019). The bioelements, the elementome, and the biogeochemical niche. Ecology, 100, e02652.
Wright, I.J., Reich, P.B., Westoby, M., Ackerly, D.D., Baruch, Z., Bongers, F., et al. (2004). The worldwide leaf economics spectrum. Nature, 428, 821–7.
Q4: Lines 84-85. Although an article is cited in relation to the chemical characteristics of the water where the mosses grew, it would be interesting if the authors provide some data of these characteristics. Perhaps average values of pH and conductivity, with maximum and minimum of the total sampling points.
R4: Following the reviewer’s suggestion, we have now added water pH and conductivity information on the springs surveyed, see L. 86-89: “In the subset of springs used here, water pH ranged from 5.17 to 8.34 (median: 7.20) and water conductivity ranged from 24 to 2094 (median: 498) µS cm-1. A detailed analysis of the water chemical composition can be found in the online material from references [25,30,31]”.
Q5: Line 92. How long were the samples immersed in acetic acid?
R5: They were submerged during 15 minutes, although some samples with strong CaCO3 incrustations needed up to an hour. We have now added this information in the method section, see L. 96-98: “Before the moss elemental analyses, we submerged all samples in a solution of acetic acid at pH 2.7 for 15 minutes in order to remove incrustations of CaCO3. A few samples required up to an hour to remove all CaCO3 incrustations”.
Q6: Figure 2. I would add x-axis data labels to the top two graphs. It is difficult to know which species each boxplot corresponds to.
R6: Thanks, we have now added the x-axis labels to the two top graphs as well (see new Figure 2).
Round 2
Reviewer 1 Report
I'm still not satisfied with replies 1 and 2. However, I leave it up to future readers to judge this article. Therefore, I suggest publishing this article in its present form.
Author Response
Q1: I'm still not satisfied with replies 1 and 2. However, I leave it up to future readers to judge this article. Therefore, I suggest publishing this article in its present form.
R1: We thank the referee for spending time reviewing our paper again. Given that the referee agrees on publishing the article in its present form, we have now checked the manuscript again and submitted it to the journal.